# A Mixed Methods Study on Community-Based Tourism as an Adaptive Response to Water Crisis in San Andrés Ixtlahuaca, Oaxaca, Mexico

**María del Rosario Reyes-Santiago** [1,*] **, Elia Méndez-García** [2] **and Patricia S. Sánchez-Medina** [2]

1 Unidad Campeche, El Colegio de la Frontera Sur (ECOSUR), Av. Rancho Polígono No. 2-A, Lerma, Campeche C.P. 24500, Mexico

2 CIIDIR-IPN Unidad Oaxaca, Instituto Politécnico Nacional, Hornos No. 1003 Santa Cruz Xoxocotlán, Oaxaca C.P. 71230, Mexico; mendezeli@hotmail.com (E.M.-G.); psanchez@ipn.mx (P.S.S.-M.)

* Correspondence: mariadel.rosario@hotmail.com

**Abstract:** Water scarcity is a threat in San Andrés Ixtlahuaca, Mexico, that imperils the survival of farming households whose food and income depend on rainfed agriculture. This research extends the framework of socioecological systems to tourism to understand how community-based tourism flourishes: not spontaneously but as part of an adaptive response to the water crisis. A research model was constructed based on mixed methods. For the qualitative approach, interviews were conducted with 12 community leaders. Results show that different capabilities have been developed throughout the adaptive cycle: information capabilities at the $\Omega$ phase; involvement capabilities at the $\alpha$ phase; self-esteem capabilities at the r phase; and resource use capabilities at the k phase. These capabilities make it possible to face the water crisis, but they also favor the implementation of tourist activity. For the quantitative approach, a questionnaire was applied to 88 community participants directly involved in tourism activities to discover the current state of the tourism-related capabilities, their shaping, and relationships. A partial least squares structural equation modeling (PLS-SEM) to test the hypotheses raised was used. The activity makes the community resilient because it seeks to conserve and improve community resources through tourism-related capabilities.

**Keywords:** adaptive cycle; capabilities; tourism; water crisis; sustainability; resilience

## 1. Introduction

Mexico is highly vulnerable to the effects of climate change. One effect of this global phenomenon is the increased variability and reduction of rainfall [1–3]. The consequent water scarcity threatens the survival of rural farming communities, whose food and income come from rainfed agriculture [4]. Therefore, analyzing the causes and consequences of this process is essential to build adaptation strategies that increase the resilience of communities and face this challenge in the best way possible [5–8].

Despite growing interest in understanding resilient systems, there is a lack of research on the factors that favor a successful transition through critical stages, based on examples of real socio-ecological systems [2,9,10]

Different capabilities that favor system resilience have been identified, especially related to information management, how information is shared and understood by the group, and the driving factors for collective action [7,11]. However, no research has been conducted on how these capabilities can be used to stimulate tourist activity.

These capabilities can be integrated as part of an adaptive capacity [12,13] or as part of a system management capability [11]. In our research, however, the capabilities are considered as different because separating them allows a better appreciation of their development over time at different stages of the adaptive cycle, to appreciate their shaping and relationships over time.

It should be noted that the adaptive cycle shows temporal changes related to the system's progression through developmental stages: rapid growth and exploitation (r); conservation (k); collapse, release, or creative destruction (Ω); and renewal or reorganization (α). Throughout these stages, various capabilities are developed [12,14].

Our research extends the framework of socioecological systems to community-based tourism. It is proposed that the community, even unintentionally, has developed tourism-related capabilities regarding the use of information, involvement, personal self-assessment, and use of resources for tourism.

Tourist activity is not separate from the dynamics of the socio-ecological system in which it operates [2,15]. In this sense, efforts for conservation and better use of natural resources also imply changes in the way in which economic activities are developed, seeking to develop sustainable and community tourism [16].

Although previous investigations were not focused on the identification of tourism capabilities, they did show signs of the construction of capabilities throughout the historical development of the system, through its adaptive cycles. These capabilities allow the community to become informed, get involved, and activate more sustainable tourism [17,18].

The development of adaptation capabilities does not work exclusively, and thus both the diversity of adaptation capabilities and the diversity of innovations they generate favor the resilience of the system in environmental terms, such as overcoming the water crisis, but also provide economic and social benefits, in the form of tourism [19].

These capabilities are built and emerge over time, responding to the characteristics and needs of each stage of the adaptive cycle. Thus, it is possible to identify them in the moment and as they develop because they are related.

This work is useful because it allows us to understand how community tourism does not happen spontaneously but is built over time, mediated by capabilities focused on the material reproduction and the symbolic community life [20]. Thus, tourism activity favors system resilience by allowing adaptation to change and uncertainty; nurturing the diversity of its economic activities; promoting the combination of different types of knowledge; and creating opportunities for self-organization [21–23].

The following section introduces the location of San Andrés Ixtlahuaca. The methodology section explains how the research model was tested using qualitative and quantitative data. At the end, the results, discussion, and conclusions are declared.

## 2. Context of San Andrés Ixtlahuaca

The municipality of San Andrés Ixtlahuaca is in the central part of the state of Oaxaca, the Central Valley region, and belongs to the Centro District. It is located southwest (SW) of the city of Oaxaca. It is bordered to the north by the municipalities of San Felipe Tejalapam and San Lorenzo Cacaotepec; to the south by San Pedro Ixtlahuaca; to the east by Santa María Atzompa; and to the west by San Felipe Tejalapam. Its approximate distance to the state capital is 10 km (see Figure 1).

San Andrés Ixtlahuaca has the history and cultural legacy of the Mixtec, Zapotec, and Mexica native people: Mixtecs from Cieneguilla, Chiyau ñu saca, Zapotecs from the sandy valley of San Andrés Ixtlahuaca Guixonachapa, and Mexicas arriving in distinct waves through the Oaxacan territory [24]. It was a place of passage for travelers and traders on the road between the Mixteca Alta and the Central Valley. In Chiyau ñu saca or Cieneguilla, there are vestiges of an altar with stone stairs, a dedicated place to stop and rest.

After the Mexican Revolution, San Andrés Ixtlahuaca progressively lost its communal territory to the benefit of actors outside the community, both private individuals and other nearby towns. This period, or rather a series of periods of agrarian conflict, ended with a resolution from the Mexican Supreme Court of Justice, which ruled in favor of San Andrés in 1965, thus ending the conflict and dropping the complaints that had been filed due to the violent events that occurred during the conflict [24]. After 1965, agricultural and commercial development brought a certain stability and prosperity to San Andrés Ixtlahuaca.

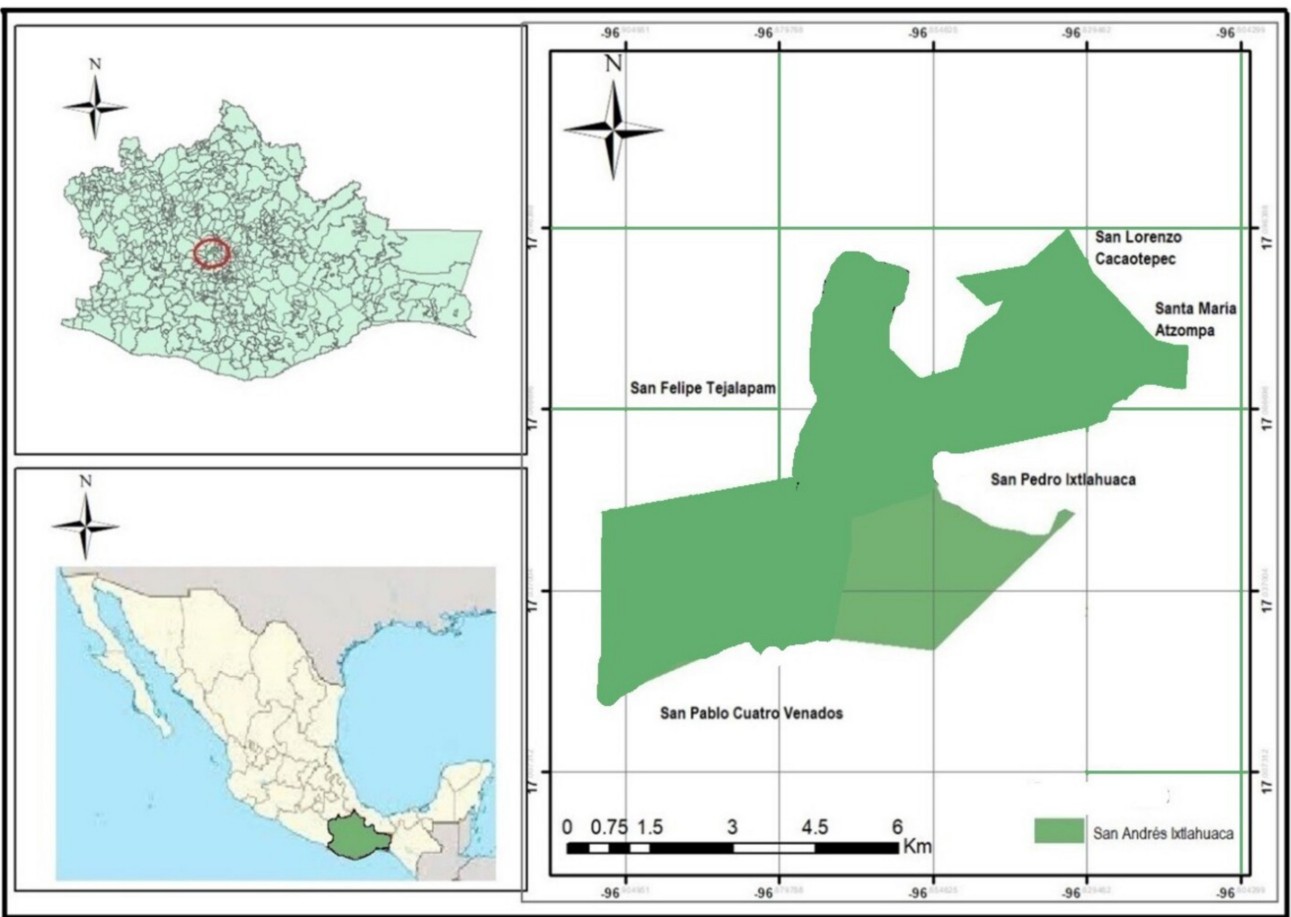

**Figure 1.** Location of San Andrés Ixtlahuaca. Source: Self-made.

Until 2000, the community had found in agriculture the source for its economic resources, and a local market niche for its products has been developed. Although the stages of adaptive cycles might be discontinuous and even overlap in time [14], in the case of San Andrés Ixtlahuaca, there is a clear before and after state caused by the water scarcity.

First, the water crisis in San Andrés Ixtlahuaca is a direct consequence of the effects of climate change, but it also has an anthropic origin. In this regard, the community has created an explanation using a story told to explain why the rains left. In summary, they believe that at a given place in the community lives the Water Serpent, and this mythical being has brought the tributaries and the harvests. However, the humans began to populate and make a lot of noise, making the water snake feel uncomfortable and leave the place. An older and respectable adult from the community also had a dream in which he saw the Water Serpent leaving the community, the crops as well as other animals that provide food to San Andrés Ixtlahuaca following it and leaving as well. Because of this deep cultural consciousness, the community understands the importance of conserving its forests and promoting a lifestyle compatible with agriculture.

This disruption has triggered the onset of a new adaptive cycle as the water scarcity is based upon a comparison scheme depending on a perspective adopted over a given time period [7].

## 3. Methodology

To show the origin, the shaping, and the relationships among the tourism-related capabilities built in San Andrés Ixtlahuaca, the following research model was developed (see Figure 2).

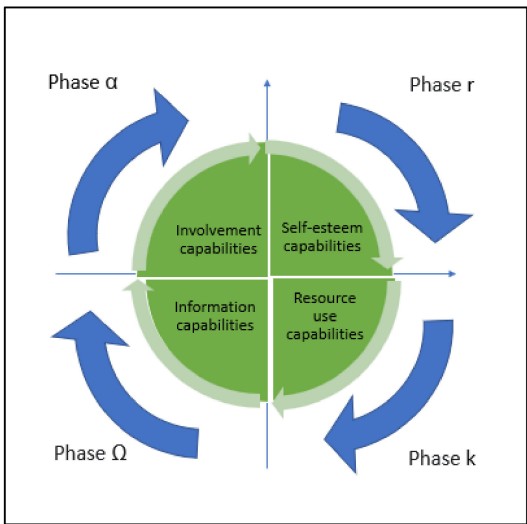

**Figure 2.** Tourism capabilities in the adaptive cycle. Note: Source: Self-made based on Gunderson and Holling (2002) and data from the community of San Andrés Ixtlahuaca, Oaxaca.

As it can be seen, as the adaptive cycle progresses, different tourism-related capabilities are developed at each stage, indicating that they are related to each other.

The Ω stage creates a perfect space for reorganization and incorporation of new models [14,25]. At this stage, it is necessary to develop information acquisition capabilities for decision making because the introduction of an inappropriate innovation may cause system collapse (Walker & Salt, 2006). Information capabilities are also necessary for protecting vital functions and prioritizing survival functions with minimal access to the needed resources [26].

In terms of tourism, the capabilities are developed for the acquisition and management of new information to the community, and they are built through various workshops, talks, and meetings with the institutions involved. However, it is also important to consider information from the past, through community remembrance. When the community notices that foreign people come to enjoy the natural landscapes, that makes it remember its ancient practice of providing lodging and food services to travelers.

The α stage prepares the environment for a new growth phase. In this stage, it is necessary to develop involvement capabilities aligned with the analysis of the causes of the problem and the generation of spaces to find and model alternatives, the capacity to rethink the working mode based on new visions, and then redefine the direction the system should follow [26].

The information handled in the previous stage has an impact on the involvement of the population in the generation and implementation of solutions, evidenced by collaboration and collective actions [7], the development of social networks, and the construction of a common vision and goals [11]. The aspects just mentioned make up the involvement capacity, allowing us to propose the following hypotheses:

**H1.** *Information capabilities have a positive and significant effect on involvement capabilities.*

An r stage system has successfully reoriented itself after the crisis and now seeks activation energy to achieve rapid growth and development. At that point, there are opportunities to test and change courses of action for growth to change [26].

It is necessary for the community population to feel capable of providing the services required by tourism [27]. This includes the self-recognition as people that possess valuable knowledge and skills applicable to tourism [28–30]. Given this, the following hypothesis is postulated:

**H2.** *Involvement capabilities have a positive and significant effect on self-esteem capabilities.*

In stage k, the potential accumulation creates a wealth increase available to the structures able to acquire, store, maintain, and use it [26]. The aim is to increase and build capabilities for the implementation of the services or to improve usage of community resources to turn them into tourist attractions [31–33].

**H3.** *Self-esteem capabilities have a positive and significant effect on resource use capabilities.*

In this research, qualitative and quantitative methodological tools were used.

Since the purpose of the research was to investigate how tourism emerges as an option for economic activity for the community, for the qualitative part, a case study was designed with a theoretical [34,35]. It was formulated for key informants, people who are knowledgeable about the subject investigated, lucid, thoughtful, and willing to speak extensively with the researcher.

Although not using statistically representative samples in qualitative research, a solid sampling strategy plays in gathering valid and reliable data to support results.

Thus, a non-probabilistic sampling design was used [36,37]. The inclusion criteria were (1) being in a position of authority at the time of conducting the study, (2) having been in a position of authority in previous years, and (3) having participated in the rescue and environmental conservation activities and tourism-targeted initiatives.

Purposive sampling was used to select respondents that were most likely to yield appropriate and useful information [38]. Using this sampling technique helps the research to gain an understanding of operations on the ground, especially from key personnel involved in community tourism activity.

It should be noted that the people interviewed can be considered the total number of key informants available to be interviewed. In this way, the twelve interviews carried out covered all community authorities involved in tourism, from 2020 to 2021, since many of them held office for more than one year and others died due to their age. Six men and six women were interviewed, aged between 38 and 73 years. The interviewees are recognized community leaders in San Andrés Ixtlahuaca, people who have held community positions such as committee members from the ejido and public offices for several terms.

The interviewees were farmers, whose main crops are tomato and pumpkin. As a complement to their agricultural activity, they are merchants and carry out other trades such as bakery and masonry. Some members have professions such as educators, architects, and biologists.

Table 1 shows the characteristics of the interviewees. It should be noted that participants had experience with the phenomenon, and they were willing to share their thoughts and memories.

During the interview, general data such as name, age, occupation, and official position were asked. Next, the conversation continued by inviting the interviewee to discuss tourism in the community. From this subject, categories arise such as crisis, change, training, involvement, and use of resources. Moreover, information on the history and major events of the community was addressed.

Based on this information from the community and the life experiences of its members, it was possible to identify that, at the beginning of the 2000s, the community of San Andrés Ixtlahuaca began an adaptive cycle triggered by water scarcity and its response to this challenge.

This information was organized through content analysis and contrasted with secondary documentary data to corroborate the information gathered from the informants. Thanks to this, the saturation point was achieved [36,39], which makes this qualitative research valid.

**Table 1.** Characteristics of the interviewees.

| | Code | Gender | Age (Years) | Community Charge | Role |
|---|---|---|---|---|---|
| 1 | RAM | Male | 42 | President of the Ejidal Commissariat of San Andrés Ixtlahuaca | Community leader, construction and agricultural activities |
| 2 | NSV | Female | 49 | Collaborator and member of the Ejidal Commissariat of San Andrés Ixtlahuaca | Agricultural activities carried out by women |
| 3 | JMJ | Female | 38 | Member of the San Andrés Ixtlahuaca Environmental Culture Center | Local vegetation conservation actions |
| 4 | AAJ | Female | 40 | President of the San Andrés Ixtlahuaca Environmental Culture Center | Development of tourist actions |
| 5 | JIJM | Male | 70 | Collaborator and member of the Ejidal Commissariat of San Andrés Ixtlahuaca | Community chronicler |
| 6 | GWSM | Male | 72 | Collaborator and member of the Ejidal Commissariat of San Andrés Ixtlahuaca | Community tour guide |
| 7 | GCRS | Male | 73 | Collaborator and member of the Ejidal Commissariat of San Andrés Ixtlahuaca | Community tour guide |
| 8 | MEVH | Female | 46 | Collaborator and member of the Ejidal Commissariat of San Andrés Ixtlahuaca | Agricultural activities carried out by women |
| 9 | PSR | Male | 55 | Collaborator and member of the Ejidal Commissariat of San Andrés Ixtlahuaca | Community connoisseur's guide to herbs and spices |
| 10 | SRH | Female | 38 | Collaborator and member of the Ejidal Commissariat of San Andrés Ixtlahuaca | Environmental monitoring activities |
| 11 | HPR | Male | 38 | Collaborator and member of the Ejidal Commissariat of San Andrés Ixtlahuaca | Environmental monitoring activities |
| 12 | MERS | Female | 58 | Collaborator and member of the Ejidal Commissariat of San Andrés Ixtlahuaca | Indigenous community cook |

Source: Self-made.

Capabilities that can be observed in each phase of adaptive cycle are shown in Table 2.

For the quantitative part, a questionnaire was designed, based on the reviewed literature and examples in the context of community tourism. The questionnaire was composed of 25 questions, and they were answered based on a five-point Likert scale.

This questionnaire was applied to 88 people from the community population involved in tourism-related activities. There is no historical data on the number of people who have been involved in tourism in San Andrés Ixtlahuaca.

It should be noted that the sample meets the criteria of a PLS-SEM sampling rule of thumb, the "10-times rule": the sample size should be greater than 10 times the maximum number of inner or outer model links pointing at any latent variable in the model [40]. With a medium effect size, statistical power of 0.8, significance level of 0.05, and three predictors, the minimum required sample size is 77 units [41,42]. Thus, the sample size achieved (N = 88) for this study is larger than required.

**Table 2.** Capabilities and adaptive cycle.

| Adaptive Cycle Phases | Capabilities |
| --- | --- |
| Ω phase | Information capabilities for acquiring information from external actors, remembering ancient knowledge, and discussing knowledge.Linking and training with agents of educational institutions, internal assemblies for dialogue, and learning from older adults in the community.Information for more efficient use of water, better agricultural practices, rainwater harvesting, and forest conservation. The community is informed and becomes aware of activities compatible with environmental care such as sustainable tourism. |
| α phase | Involvement capabilities to find alternatives to the model that is already outdated. Driving of decision-making practices for the reinforcement of the communal ownership of the land, to go down a path of environmental conservation.Community organization in a committee to build the center of the environmental culture of said community. |
| r phase | Self-esteem capabilities: the local population values their knowledge and skills and puts them to work for community development, implementing greenhouse and water harvesting systems; community tourism guides who are knowledgeable about environmental issues are trained. |
| k phase | Resource use capabilities: In order to prepare for a greater accumulation of potential, the diversification of activities is the way to do it. The benefits obtained from environmental care are diversified, as payment is obtained for environmental services and financing, in turn, increasing the volume and quality of the agricultural production. The first tourism activities are carried out in the environmental culture center, and a work season begins with tourism experts helping to improve the activities. |

Source: Self-made.

Regarding the characteristics of our sample, people between 15 and 64 years old participated, 41 men and 47 women, with an educational level ranging from secondary to postgraduate.

With the data obtained, a partial least squares structural equation modeling (PLS-SEM) analysis was performed using the Smart PLS software. The information obtained from the questionnaires shows the shaping and relationships of the actual tourism-related capabilities.

## 4. Results and Discussion

The results obtained from both the quantitative and qualitative tools are presented below.

### 4.1. Qualitative Data

#### 4.1.1. Ω Stage

Around the year 2000, water shortages disrupted the community's way of life economically, socially, and culturally. In this regard, a prompt release of accumulated resources and biomass could be observed, corresponding to the destructive phases of the cycle [14,25,43].

The lost resources included crops, as well as the depletion of food reserves, objects, and money that the farming families had accumulated. Actions were taken in response to the negative social and economic effects of the water crisis, such as emigration in search of work, as well as the overexploitation of the forest, used to produce charcoal. These actions explain why the community favored the continuation of agricultural activity. Moreover, the first dams and dykes were built, and the reforestation work began, using the community's materials and knowledge.

There was also a need for resources and information that prompted links with agents such as various governmental institutions, e.g., National Environmental Commission

(CONAM), National Forestry Commission (CONAFOR), Risk Sharing Trust (FIRCO), Ministry of Agriculture, Livestock, Rural Development, Fisheries and Food (SAGARPA), as well as educational institutions such as the Interdisciplinary Research Center for Integral Regional Development (CIIDIR) and the Technological Institute of Oaxaca (ITO).

In general, these links provided access to key information about water use, rainwater capture and storage, and better agricultural and forestry practices, as well as access to financial support. In particular, the community acquired information about the implementation of tourism in places where the environment is well preserved.

### 4.1.2. α Stage (2002)

In the face of water scarcity, it is necessary to create spaces that favor the participation of the different stakeholders in the decision-making processes, as well as to achieve constant communication between the different levels of the processes and to generate methods of conflict resolution, through institutions that have the flexibility to face diverse situations [7].

When the community of San Andrés Ixtlahuaca learned that its natural environment is fundamental to its survival, it became necessary to decide on a future-facing course of action. This was possible through the community assembly, which reached a common consensus. In this way, community members became involved in the proposed actions.

It should be noted that the decisions made in the community assemblies were carried out at different times and at different levels. Usually, the topics are first proposed in the ejido assemblies, and then they go to the general assemblies. Outside actors are not welcome, which makes information from members and key stakeholders vital to understanding community processes.

The community decides to preserve their traditional community governance and collective land tenure. In this way, it is possible to observe the establishment of a joint vision materialized in a Master Plan for the care of the community's natural resources.

At this stage, population involvement can also be observed through the generation of a labor organization, which will also be used for the operation of tourism-related activities. Regarding tourism, in the beginning, training was provided at an ecotourism center in San Juan Nuevo, Michoacán. As a result of this experience, the participants realized that they are also capable of developing tourist activities, and a volunteer committee was formed to work in an area identified as having potential for tourism.

### 4.1.3. r Stage (2010)

The diversification of productive activities through the integration of complementary options compatible with agricultural activity was attainable at this moment in San Andrés Ixtlahuaca's history. Construction at the San Andrés Ixtlahuaca Environmental Culture Center began. Work was done on its planning and construction and in the creation of a team to offer tours, starting with groups from local schools and then expanding to tourists.

It should be noted that there has been a development in people's self-esteem as the community members perceive themselves as capable of creating communication and collaborative links with different actors, recognizing their potential and ability to integrate new service activities, which demand skills and attitudes different from those developed in traditional agricultural activity.

### 4.1.4. k Stage (2013)

The community of San Andrés Ixtlahuaca continued to work on building infrastructure for tourism such as the community kitchen and the zip line. National Autonomous University of Mexico (UNAM) students are currently carrying out, throughout the community, the identification of the tourist attractions of San Andrés Ixtlahuaca. This project aims to revalue its resources, first by the community members and, later, by proudly showing them abroad. This exercise is important because it is the basis for developing a successful community and sustainable tourist product.

*4.2. Quantitative Data*

As it can be seen, throughout the stages of the adaptive cycle, tourism-related capabilities have been developed. Their shaping and relationships are shown below.

To assess the measurement model, it is necessary to evaluate two other models: the reflective measurement model and the formative measurement model. The following table shows the variables, the indicators that comprise them, and whether they are formative or reflective constructs (see Table 3).

**Table 3.** Variables and indicators.

| Construct/Indicator | Variable | Reflective/Formative |
|---|---|---|
| INF1. Attendance at training workshops<br>INF2. Attendance of information talks<br>INF3. Attendance at fairs and exhibitions | Information capabilities | Formative |
| INV1. Knowledge about initiatives and projects inside community<br>INV2. Participation in initiatives and projects inside community<br>INV3. Knowledge about initiatives and projects outside community<br>INV4. Participation in initiatives and projects outside community | Involvement capabilities | Reflective |
| SC1. Interaction between local residents and tourists<br>SC2. Provide important opinions for community tourism<br>SC3. Provide important skills for community tourism<br>SC4. Provide valuable experiences for community tourism<br>SC5. Identification resources for tourism | Self-esteem capabilities | Reflective |
| URC1. Safeguard and promote oral stories, legends, and traditions of the region<br>URC2. Safeguard and promote gastronomy and ingredients of the region<br>URC3. Safeguard and promote plants and animals of the region<br>URC4. Safeguard and promote rivers, mountains, and trails in the region | Resource use capabilities | Reflective |

Source: Self-made.

In this research, the variable "face-to-face forms of learning" is considered as a formative variable because the items that comprise it are causes or antecedents of the construct [44,45]. The three items that comprise it refer to how the population has been able to access tourism-related knowledge through training workshops, talks, and tourism fairs (INF1, INF2, and INF3).

4.2.1. Evaluation of the Formative Measurement Model

In the formative model, each indicator represents a dimension of the meaning of the latent variable. Eliminating an indicator means that the variable loses part of its meaning, hence the importance of the indicators causing the construct. For the evaluation of the training model, three elements were used: weights, loads, and collinearity of the indicators [46].

The evaluation of the formative measurement model is integrated with the evaluation of the weight and loads of the indicators, as well as the collinearity between them due to their loadings being greater than 0.5 and their acceptable variance inflation factor (VIF) values of less than 3 [46]. The items shown in Table 4 were kept.

**Table 4.** Formative measurement model.

| Variable | Items | Weight | Load | VIF |
|----------|-------|--------|------|-----|
| Information capabilities | INF1. Attendance at training workshops | 0.636 | 0.266 * | 1.686 |
| | INF2. Attendance of information talks | 0.739 | 0.113 * | 2.179 |
| | INF3. Attendance at fairs and exhibitions | 0.952 | 0.785 *** | 1.509 |

\* $p \le 0.1$; \*\*\* $p \le 0.001$. Source: Self-made.

### 4.2.2. Evaluation of the Reflective Measurement Model

For the assessment of the reflective measurement model, construct reliability, convergent validity, and discriminant validity were considered.

The SmartPLS software provides the composite reliability index, Cronbach's alpha, and Dijkstra–Henseler Value. Values greater than 0.8 are adequate for strict reliability [47–50]. As it is shown in Table 5, the research values are greater than 0.8 in all cases.

Convergent validity is considered as the degree to which a measure correlates positively with other measures from the same construct. It implies that a set of indicators represents a single construct [47,49]. This aspect is validated by verifying that the factorial loads of the indicators are higher than the minimum threshold of 0.7 [50]. This is the case in the model presented here (see Table 3). An AVE (average variance extracted) value greater than 0.5 indicates convergent validity.

Discriminant validity is assessed using cross-loading analysis, the Fornell–Larcker criterion, and the HTMT (heterotrait-monotrait ratio).

The external loading of an indicator associated with a variable must be greater than its other cross-loadings (its correlations) on the other variables in the model. In this way, it can be verified that no item is loaded with greater intensity on any construct other than the one it measures. The evaluation and reporting of the cross-loadings are done using a table with rows for the indicators and columns for the variables. This condition was met in the data presented in Table 6.

The Fornell–Larcker criterion compares the square root of the variance values extracted from the average (AVE) with the correlations of the latent variables. It verifies that the square root of the variance extracted from the mean of a variable is greater than the correlation of this variable with another variable [51,52]. This criterion was met as it can be seen in Table 7.

The final evaluation criterion of the reflective model is the calculation of HTMT. In a well-fitted model, the HTMT ratio should be significantly less than 1 (see Table 8).

Several aspects were considered for the assessment of the structural model. When assessing collinearity between constructs, values were found between 1 and 1.27 VIF. Values of VIF less than 3 are acceptable [48]; therefore this aspect is met.

The adjusted value for the coefficient of determination ($R^2$) was calculated. The interpretation of this indicator is analogous to a regression: it represents the combined effects of the exogenous latent variables on the endogenous latent variable. Values between 0.1 and 0.25 indicate a weak explanatory power, values under 0.5 are considered moderate, and values between 0.5 and 0.75 are strong [47,53].

The predictive relevance of the model was also calculated using the ($Q^2$) statistics, which measure the predictability of the data observed through the routing model. Values below 0.25 indicate small predictive accuracy, values between 0.25 and 0.5, medium accuracy, and values greater than 0.5, large accuracy [42,48].

**Table 5.** Reflective measurement model.

| Variable | Indicator/Item | Load | Cronbach's Alpha | Dijistra–Henseler (rho_A) | AVE | Composite Reliability |
|---|---|---|---|---|---|---|
| Involvement capabilities | INV1. Knowledge about initiatives and projects inside community | 0.728 *** | 0.837 | 0.851 | 0.674 | 0.890 |
| | INV2. Participation in initiatives and projects inside community | 0.887 *** | | | | |
| | INV3. Knowledge about initiatives and projects outside community | 0.870 *** | | | | |
| | INV4. Participation in initiatives and projects outside community | 0.800 *** | | | | |
| Resource use capabilities | URC1. Safeguard and promote oral stories, legends, and traditions of the region | 0.749 *** | 0.832 | 0.855 | 0.663 | 0.885 |
| | URC2. Safeguard and promote gastronomy and ingredients of the region | 0.871 *** | | | | |
| | URC3. Safeguard and promote plants and animals of the region | 0.841 *** | | | | |
| | URC4. Safeguard and promote rivers, mountains, and trails in the region | 0.807 *** | | | | |
| Self-esteem capabilities | SC1. Interaction between local residents and tourists | 0.764 *** | 0.874 | 0.911 | 0.672 | 0.910 |
| | SC2. Provide important opinions for community tourism | 0.842 *** | | | | |
| | SC3. Provide important skills for community tourism | 0.903 *** | | | | |
| | SC4. Provide valuable experiences for community tourism | 0.835 *** | | | | |
| | SC5. Identification resources for tourism | 0.747 *** | | | | |

*** $p \leq 0.001$. Source: Self-made.

As it can be seen in Table 9, the explanatory power for involvement ($R^2 = 0.113$) and community resources ($R^2 = 0.161$) is low, while the explanatory power for self-esteem ($R^2 = 0.394$) is moderate. All values for $Q^2$ correspond to a small predictive accuracy.

$f^2$ indicates the degree to which an exogenous construct explains an endogenous construct in terms of $R^2$: $0.02 < f^2 < 0.15$ for small effect, $0.15 < f^2 < 0.35$ for moderate effect, and $f^2 > 0.35$ for large effect. Accordingly, as shown in Table 10, the effects are moderate.

For the assessment of the overall model, bootstrap-based exact fit tests with 5000 subsamples were calculated to determine the probability of obtaining a discrepancy between the correlation matrix implied by the model and the empirical correlation matrix. The values obtained must be less than the bootstrap values at 95% or 99%, so much for SRMR (standardized root mean square residual), dULS (unweighted least squares discrepancy), and dG (geodesic discrepancy) [53]. As it can be seen here, these criteria are met.

Information capabilities has a direct positive effect on involvement capabilities ($\beta = 0.336$); H1 is supported. Involvement capabilities has a direct positive effect on self-esteem capabilities ($\beta = 0.243$); H2 is supported. In terms of this research, self-worth is understood as feeling comfortable interacting with tourists; this is explained because, given the information received, the population feels able to contribute their valuable skills and ideas to tourism. In addition, self-esteem capabilities has a direct positive effect on resource use capabilities ($\beta = 0.0401$); therefore H3 is accepted. In the beginning, the focus of the project was solely on natural resources, but the community is now rediscovering the value of other resources, especially cultural ones.

**Table 6.** Cross-loadings.

| Variable | Indicator/Item | Load | | |
|---|---|---|---|---|
| Involvement capabilities | INV1. Knowledge about initiatives and projects inside community | **0.728** | 0.404 | 0.332 |
| | INV2. Participation in initiatives and projects inside community | **0.887** | 0.300 | 0.382 |
| | INV3. Knowledge about initiatives and projects outside community | **0.870** | 0.272 | 0.326 |
| | INV4. Participation in initiatives and projects outside community | **0.800** | 0.229 | 0.319 |
| Resource use capabilities | URC1. Safeguard and promote oral stories, legends, and traditions of the region | 0.261 | **0.749** | 0.251 |
| | URC2. Safeguard and promote gastronomy and ingredients of the region | 0.271 | **0.871** | 0.385 |
| | URC3. Safeguard and promote plants and animals of the region | 0.294 | **0.841** | 0.300 |
| | URC4. Safeguard and promote rivers, mountains, and trails in the region | 0.342 | **0.807** | 0.352 |
| Self-esteem capabilities | SC1. Interaction between local residents and tourists | 0.287 | 0.371 | **0.764** |
| | SC2. Provide important opinions for community tourism | 0.296 | 0.339 | **0.842** |
| | SC3. Provide important skills for community tourism | 0.38 | 0.292 | **0.903** |
| | SC4. Provide valuable experiences for community tourism | 0.413 | 0.376 | **0.835** |
| | SC5. Identification resources for tourism | 0.306 | 0.265 | **0.747** |

Source: Self-made.

**Table 7.** Fornell–Larcker criterion.

| | Self-Esteem Capabilities | Resource Use Capabilities | Involvement Capabilities |
|---|---|---|---|
| Self-esteem capabilities | **0.824** | | |
| Resource use capabilities | 0.357 | **0.818** | |
| Involvement capabilities | 0.412 | 0.401 | **0.820** |

Source: Self-made.

**Table 8.** HTMT.

| | Involvement Capabilities | Resource Use Capabilities |
|---|---|---|
| Resource use capabilities | 0.436 | |
| Self-esteem capabilities | 0.480 | 0.460 |

Source: Self-made.

**Table 9.** $R^2$ and $Q^2$ of the structural model.

| | $R^2$ | $Q^2$ |
|---|---|---|
| Involvement capabilities | **0.113** | 0.063 |
| Resource use capabilities | 0.161 | 0.091 |
| Self-esteem capabilities | 0.394 | **0.236** |

Source: Self-made.

**Table 10.** Path coefficients.

| | Path Coefficient | Standard Deviation | T Statistics | *p* Values | f² |
|---|---|---|---|---|---|
| Involvement capabilities → Self-esteem capabilities | 0.243 | 0.088 | 2.756 | 0.006 | 0.113 |
| Self-esteem capabilities → Resource use capabilities | 0.401 | 0.084 | 4.777 | 0.000 | 0.192 |
| Information capabilities → Involvement capabilities | 0.336 | 0.102 | 3.286 | 0.001 | 0.127 |
| Information capabilities → Self-esteem capabilities | 0.503 | 0.088 | 5.724 | 0.000 | 0.370 |

Source: Self-made.

## 5. Conclusions

Communities are sociological systems that go through adaptive cycles, during which construction and destruction stages can be observed.

San Andrés Ixtlahuaca can be identified as a socioecological system characterized by a particular way of inhabiting the environment. This understanding of the reality is based on an ancestral tradition that is expressed in current elements, such as social relations and practices, sense of territory, management criteria, and the defense of its resources.

The community adopted a position on the problem at hand according to two aspects: first, how they perceive the impact of disasters or threats from the environment; second, the access to resources to generate and implement strategies to deal with them.

The introduction and implementation of new activities can be better understood by considering the historical moment that the community is going through at any time as each stage's characteristics enable the development of capabilities for new activities, such as tourism. In the present work, we deepen the knowledge about how tourism is part of an adaptive response of the population, the water crisis in the case of San Andrés Ixtlahuaca, and can be treated as a multi-functionality option for the reproduction of the material and symbolic life of the community.

This research is useful because it allows us to understand how community tourism does not arise spontaneously but is developed over time and is mediated by capabilities aimed at the reproduction of the material and symbolic life of the community. Thus, tourism-related activity favors system resilience by allowing adaptation to change and uncertainty, nurturing the diversity of its economic activities, promoting the combination of different types of knowledge, and creating opportunities for self-organization.

Moreover, it was shown that tourist activity does not appear spontaneously, but rather it is developed over time, using different learning processes in the community, which have not been directly focused on tourism but allow the initiative to originate from the community itself.

However, the capabilities observed need to be strengthened. In this regard, the challenges are to provide ongoing specialized training in tourism, involving more local people and visitors in the activities, strengthening the local leadership, and integrating cultural resources into the tourism-related services.

Finally, although the community has been able to cope with the water crisis, it is necessary to strengthen economic activities, especially tourism [54], so they may have the resources needed to respond to the new threats presented by the environment, which will create new adaptive cycles within the community.

**Author Contributions:** Methodology, E.M.-G.; Supervision, P.S.S.-M.; Writing—original draft, M.d.R.R.-S. All authors have read and agreed to the published version of the manuscript.

**Funding:** This research was funded by: Postdoctoral Program for Indigenous Mexican Women in Science, Technology, Engineering and Mathematics of Mexican Council of Science and Technology (CONACYT, in Spanish acronym)—Center for Research and Higher Studies in Social Anthropology (CIESAS, in Spanish acronym)—International Development Research Center Canada (IDRC).

**Data Availability Statement:** Not applicable.

**Acknowledgments:** We would like to thank Postdoctoral Program for Indigenous Mexican Women in Science, Technology, Engineering and Mathematics of Mexican Council of Science and Technology (CONACYT, in Spanish acronym)—Center for Research and Higher Studies in Social Anthropology (CIESAS, in Spanish acronym)—International Development Research Center Canada (IDRC); for assisting to finalize this paper.

**Conflicts of Interest:** The authors declare no conflict of interest.

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
