# Peer review of "A Mixed Methods Study on Community-Based Tourism as an Adaptive Response to Water Crisis in San Andrés Ixtlahuaca, Oaxaca, Mexico"

_sustainability, doi:10.3390/su14105933_

Round 1

Reviewer 1 Report

Reviewers' comments:

I have received the manuscript Sustainability-1607330 for reviewing. The structure and logic of this manuscript are very chaotic. The method has huge shortcomings and deficiencies.  I would like to give my specific comments as following:

  1. The abstract should include the background, significance, objective, method and result of the study. However, this study has few descriptions of results. You should rewrite the abstract based on the results.
  2. Lines 36-40 and Lines 56-80. These sentences are the content and significance of this study. There are too many such contents in the introduction.
  3. Line 88 and 120. The historical characteristics of San Andrés Ixtlahuaca may be useful for this study. However, you should provide a location map of San Andrés Ixtlahuaca because many readers may not know where San Andrés Ixtlahuaca is.
  4. Line 121 to 226. I really cannot understand the existence significance of this section. This section looks like an adaptive cycle and capability’s introduction about San Andrés Ixtlahuaca. You could refine the content of this section. Then, you move this section to the introduction.
  5. I think there are great deficiencies and shortages in the method section. You do not explain the basic information of interviews (12 direct and 88 face-to-face interviews), such as gender, age, occupation, and income. In particular, the proportion of the sample to the total population of the region. The basic information of interviews also impacts interview information. Then, interview information also impacts the result of this study. Therefore, you should provide more detailed information about interviews to prove the reliability and representativeness of interview information.
  6. Abbreviations are usually defined when they first appear in the main text and are used in the remainder of the manuscript.
  7. Line 266 to 275. These paragraphs look like introduce the method. Therefore, I really confused why you write them here?
  8. Only the results of this study need to be stated in the conclusion, and there is no need to cite other people's literature.

Author Response

Response to Reviewer 1 Comments

  1. Point1. The abstract should include the background, significance, objective, method and result of the study. However, this study has few descriptions of results. You should rewrite the abstract based on the results.

Response 1. The summary has been improved, and the research results were highlighted.

  1. Point 2. Lines 36-40 and Lines 56-80. These sentences are the content and significance of this study. There are too many such contents in the introduction.

Response 2. Redundant elements in the introduction have been removed.

  1. Point 3. Line 88 and 120. The historical characteristics of San Andrés Ixtlahuaca may be useful for this study. However, you should provide a location map of San Andrés Ixtlahuaca because many readers may not know where San Andrés Ixtlahuaca is.

Response 3. A community location map was added.

  1. Point 4. Line 121 to 226. I really cannot understand the existence significance of this section. This section looks like an adaptive cycle and capability’s introduction about San Andrés Ixtlahuaca. You could refine the content of this section. Then, you move this section to the introduction.

Response 4. This section has been eliminated, and the key elements have been relocated to the introduction and methodology sections.

  1. Point 5. I think there are great deficiencies and shortages in the method section. You do not explain the basic information of interviews (12 direct and 88 face-to-face interviews), such as gender, age, occupation, and income. In particular, the proportion of the sample to the total population of the region. The basic information of interviews also impacts interview information. Then, interview information also impacts the result of this study. Therefore, you should provide more detailed information about interviews to prove the reliability and representativeness of interview information.

Response 5. More information is presented for greater clarity and reliability of our sample.

  1. Point 6. Abbreviations are usually defined when they first appear in the main text and are used in the remainder of the manuscript.

Response 6. This observation has been heeded; however, some terms have been kept alongside their acronyms, even after they have already been used once due to their relevance in the study’s context.

  1. Point 7. Line 266 to 275. These paragraphs look like introduce the method. Therefore, I really confused why you write them here?

Response 7. This information has been relocated to the methodology section.

  1. Only the results of this study need to be stated in the conclusion, and there is no need to cite other people's literature

Response 8. References to other authors in the conclusion section have been removed.

Reviewer 2 Report

Dear authors,

The article deals with a highly relevant topic because it addresses the vulnerability of populations to extreme weather events and the ability to react and be resilient. Congratulations.

I am not an expert in statistics, but I understand that the work is relevant and can contribute to analysing cases like this.

I will divide my observations into two parts:

  1. AS TO SHAPE:

(a) the words "capability" and "capabilities" are overused throughout the text (66 times in 13 pages);

(b) many of the acronyms that appear throughout the text are not defined (VIF or FIV?, p, among others)

(c) The title does not reflect the article's central theme, which is "the use of a statistical model" and not "a community-based tourism".

  1. AS TO THE CONTENT:

(a) An essential information for the reader was not presented in the text. It is the reason (or reasons) that generated the water crisis in the area. Climate changes? Anthropic action?

(b) The interview script and the questionnaire could be presented in the article so that the reader can understand the data that supplied the model.

(c) In the "Results and discussion" item  – the authors dedicated themselves to presenting the modelling results, pointing out at the end that the hypotheses raised were validated using the method. Reproducibility of the methodology is a fundamental value in a methodological article such as this. A review of the "Methodology" and "Results & Discussion" items is suggested to provide the readers with a better understanding of the results.

Best regards.

Author Response

Response to Reviewer 2 Comments

 Point 1. The words "capability" and "capabilities" are overused throughout the text (66 times in 13 pages);

Response 1. The wording has been improved; however, in most cases, the redundancy is preferred to avoid changing the meaning of the text.

Point 2. Many of the acronyms that appear throughout the text are not defined (VIF or FIV?, p, among others)

Response 2. Acronyms have been defined, and explicitly corrected, in the text.

Point 3.  The title does not reflect the article's central theme, which is "the use of a statistical model" and not "a community-based tourism".

Response 3. Title has been changed: “Mixed methods study on community-based tourism as an adaptive response to the water crisis in San Andrés Ixtlahuaca, Oaxaca, Mexico”.

Point 4. An essential information for the reader was not presented in the text. It is the reason (or reasons) that generated the water crisis in the area. Climate changes? Anthropic action?

Response 4. This has been reviewed and the information about how the community explains the water scarcity is detailed in the context section.

Point 5.  The interview script and the questionnaire could be presented in the article so that the reader can understand the data that supplied the model.

Response 5. More information about the interview and the questionnaire used is presented. The interview was open-ended, and the questionnaire contents can be found in Table 1. Variables and indicators.

Point 6.  In the "Results and discussion" item  – the authors dedicated themselves to presenting the modelling results, pointing out at the end that the hypotheses raised were validated using the method. Reproducibility of the methodology is a fundamental value in a methodological article such as this. A review of the "Methodology" and "Results & Discussion" items is suggested to provide the readers with a better understanding of the result.

Response 6. The methodology and results sections have been reshaped accordingly to this observation.

In general, the document has been restructured based on the recommendations of the reviewer. Finally, proofreading of the English language has been performed.

Reviewer 3 Report

The paper deals with a very important topic and thematically is well-targeted and appropriate to the journal scope. Results and discussion needs to be improved.

Author Response

Response to Reviewer 3 Comments

Point 1. The paper deals with a very important topic and thematically is well-targeted and appropriate to the journal scope. Results and discussion needs to be improved

Response 1. The results and discussion section has been reviewed and improved.

In general, the document has been restructured based on the recommendations of the reviewer. Finally, proofreading of the English language has been performed.

Reviewer 4 Report

Thank you for the article. I have some specific comments below:

In the abstract, I would include the location (Mexico) in the first sentence, as the statement "Water scarcity is a threat that imperils the survival of farming communities, whose food and income depend on rainfed agriculture" could be misleading to a global audience (e.g., in many parts of the United States, agriculture is not rainfed). Otherwise, your abstract is well-written. 

Lines 41-50: Can you expand on these ideas just a bit more? More fully develop the idea of the socioecological systems approach. I assume you will go into more detail in the methods, but even in the intro a little more detail would be good. 

"The adaptive cycle" is mentioned. Is this well understood in the literature? A broader audience might not be aware of the specifics in the adaptive cycle. I suggest explaining the cycle in 1-3 sentences in the intro. If the cycle is specific to your work, please make that clear as well. I see later in the intro you mention that the cycle will be covered, but I still believe earlier clarification is needed. 

I got to section 3 and there the adaptive cycle is explained, and matches well with literature/common adaptive cycle ideas. Perhaps including the citation earlier would clear up the confusion I had in the intro. 

Line 172-174: Are you able to elaborate on the community assembly? What did it look like? Was it a series of meetings? 

Line 250-251: Is the word "times" supposed to be doubled as such?

There's a lot about the stats of the methods, but nothing mentioned the types of questions included in the questionnaire. 

Overall takeaways: I feel as though the paper is lacking sufficient detail on what actually happened in the community. The adaptive cycle you discuss is taken from the 3,000 feet view (so to speak) or rather lacks any details, and I don't get a real sense of what actually happened in the community. For publication, the paper needs to be greatly expanded on the specific details of what happened in the community, as well as some of the details of the surveys, etc. The "why should we care" and "what happened" are both significantly missing from the work. I am hopeful you have enough information to address these concerns, as the rest of the paper is well-written and seems to follow a logical model for the adaptive cycle. 

Author Response

Response to Reviewer 4 Comments

Point 1. In the abstract, I would include the location (Mexico) in the first sentence, as the statement "Water scarcity is a threat that imperils the survival of farming communities, whose food and income depend on rainfed agriculture" could be misleading to a global audience (e.g., in many parts of the United States, agriculture is not rainfed). Otherwise, your abstract is well-written. 

Response 1.The location (Mexico) where the research was done has been included in the abstract.

Pont 2. Lines 41-50: Can you expand on these ideas just a bit more? More fully develop the idea of the socioecological systems approach. I assume you will go into more detail in the methods, but even in the intro a little more detail would be good. 

"The adaptive cycle" is mentioned. Is this well understood in the literature? A broader audience might not be aware of the specifics in the adaptive cycle. I suggest explaining the cycle in 1-3 sentences in the intro. If the cycle is specific to your work, please make that clear as well. I see later in the intro you mention that the cycle will be covered, but I still believe earlier clarification is needed. 

I got to section 3 and there the adaptive cycle is explained, and matches well with literature/common adaptive cycle ideas. Perhaps including the citation earlier would clear up the confusion I had in the intro. 

Response 2. Information about the adaptive cycle is added in the introduction. Subsequently, the methodology delves into how the concepts are made operational.

Point 3. Line 172-174: Are you able to elaborate on the community assembly? What did it look like? Was it a series of meetings? 

Response 3. More information about the community assembly has been added in the Results and discussion section.

Point 4. Line 250-251: Is the word "times" supposed to be doubled as such?

Response 4. The wording of this paragraph has been improved for better understanding.

Point 5. There's a lot about the stats of the methods, but nothing mentioned the types of questions included in the questionnaire. 

Response 5. More information about the questionnaire is presented. The items that were taken in consideration can be seen in Table 1. Variables and indicators. .

Point 6. Overall takeaways: I feel as though the paper is lacking sufficient detail on what actually happened in the community. The adaptive cycle you discuss is taken from the 3,000 feet view (so to speak) or rather lacks any details, and I don't get a real sense of what actually happened in the community. For publication, the paper needs to be greatly expanded on the specific details of what happened in the community, as well as some of the details of the surveys, etc. The "why should we care" and "what happened" are both significantly missing from the work. I am hopeful you have enough

Response 6. Descriptive elements of the community processes in which specific capabilities for tourism are developed from the perspective of key informants have been incorporated.

In general, the document has been restructured based on the recommendations of the reviewer. Finally, proofreading of the English language has been performed.

Round 2

Reviewer 1 Report

Although the author has answered our questions, the author has not introduced the "interview information" in detail. What percentage of the total population of the region is the sample? What is the content of the interview? Whether the content of the interview with each person is consistent?I will consider this article for acceptance only if the author has explained these issues well.

Author Response

Response to Reviewer 1(Round 2) Comments

Point 1. Although the author has answered our questions, the author has not introduced the "interview information" in detail. What percentage of the total population of the region is the sample? What is the content of the interview? Whether the content of the interview with each person is consistent?I will consider this article for acceptance only if the author has explained these issues well.

Response 1. The answer was included in the methodology section of the research paper:

Since the purpose of the research was to investigate how tourism emerges as an option for economic activity for the community, for qualitative part, a case study was designed with a theoretical sample (Hammersley & Atkinson, 2001; Guber, 2011; Mendieta Izquierdo, 2015). It was formulated for key informants, people who are knowledgeable about the subject investigated, lucid, thoughtful, and willing to speak extensively with the researcher.

Thus, a non-probabilistic sampling design was used (Denzin & Lincoln, 2017; Parra, 2019). The inclusion criteria were 1) being in a position of authority at the time of conducting the study, 2) having been in a position of authority in previous years, and 3) having participated in rescue and environmental conservation activities and tourism-targeted initiatives.

In this way, the twelve interviews carried out covered all community authorities involved in tourism, from 2020 to 2021, since many of them held office for more than one year and others died due to their age.

During the interview, general data such as name, age, occupation, and official position were asked. Next, the conversation continued by inviting the interviewee to discuss tourism in the community. From this subject, categories arise such as crisis, change, training, involvement, and use of resources. Also, information on the history and major events of the community was addressed.

This information was organized through content analysis and contrasted with secondary documentary data to corroborate the information gathered from the informants. Thanks to this, the saturation point was achieved (Denzin & Lincoln, 2017; Ortega-Bastidas, 2020), which makes this qualitative research valid.

Reviewer 2 Report

I consider that the article has been sufficiently improved.

Author Response

Response to Reviewer 2 (Round 2) Comments

Point 1. I consider that the article has been sufficiently improved.

Response 1. Thanks so much for your feedback.

Round 3

Reviewer 1 Report

I still think the ‘interview information’ is insufficient.

Author Response

Response to Reviewer 1(Round 3) Comments

Point 1: I still think the ‘interview information’ is insufficient.

Response 1: The information requested by the reviewer in the previous round was added to the text. In this round, the reviewer does not detail what is the missing information. However, in general, the text has been improved, according to the reviewer's recommendations.